# Multipronged Approach to Profiling Metabolites in *Beta vulgaris* L. Dried Pulp Extracts Using Chromatography, NMR and Other Spectroscopy Methods

**DOI:** 10.3390/foods12183510

**Published:** 2023-09-21

**Authors:** Joshua Fiadorwu, Kiran Subedi, Daniel Todd, Mufeed M. Basti

**Affiliations:** 1Department of Applied Science and Technology, College of Science and Technology, North Carolina Agricultural and Technical State University, Greensboro, NC 27411, USA; jefiadorwu@aggies.ncat.edu; 2Analytical Services Laboratory, College of Agriculture and Environmental Sciences (CAES), North Carolina Agricultural and Technical State University, Greensboro, NC 27411, USA; ksubedi@ncat.edu; 3Triad Mass Spectrometry Facility, University of North Carolina at Greensboro, Greensboro, NC 27412, USA; todd.da@pg.com

**Keywords:** beetroot (*Beta vulgaris* L.), Total Correlation Spectroscopy (TOCSY), Gas Chromatography-Mass Spectrometry (GC-MS), selective TOCSY, homonuclear single quantum correlation (HSQC), fatty acid methyl esters (FAME)

## Abstract

Beetroot (*Beta vulgaris* L.) is known for being a rich source of phytochemicals, minerals and vitamins. This study aims to show how the combination of extraction/chromatography/mass spectrometry and NMR offers an efficient way to profile metabolites in the extracts of beetroot. Such combination may lead to the identification of more nutritional or medicinal compounds in natural products, and it is essential for our ongoing investigation to study the selective adsorption/desorption of these metabolites’ on/off nanoparticles. The aqueous and organic extracts underwent analyses using UV-vis spectroscopy; GC-MS; LC-MS; ^1^H, ^13^C, ^31^P, TOCSY, HSQC, and selective TOCSY NMR experiments. Polar Extract: The two forms of betalain pigment were identified by UV-vis and LC MS. Fourteen amino acids, sucrose, and other compounds, among which is riboflavin, were identified by LC-MS. Two-dimensional TOCSY showed the spin coupling correlations corresponding to some of these compounds. The HSQC spectrum showed ^1^H/^13^C spin correlation in sucrose, confirming its high abundance in beetroot. Organic Extract: GC-MS data enabled the identification of several compounds including six fatty acid methyl esters (FAME) with higher than, on average, 90% similarity score. Selective TOCSY NMR data showed the spin coupling pattern corresponding to oleic, linoleic, and linolenic fatty acids. ^31^P NMR spectra indicate that phospholipids exist in both the organic and aqueous phase.

## 1. Introduction

Red beet or beetroot (*Beta vulgaris* L.) is a root vegetable in the Chenopodiaceae family [1,2] which is now cultivated throughout the world. The main components in beets are water (87.57%), carbohydrates (9.56%), protein (1.61%) and lipids (0.17%) [3]. It is also a great source of micronutrients, including minerals such as copper, iron, manganese, sodium, calcium, magnesium, potassium, phosphorus, and selenium [4]; and vitamins such as ascorbic acid ©, choline (vitamin B4), riboflavin, and niacin (vitamin B3) [5,6] as well as dietary nitrates [7]. Additionally, beets contain phytochemicals, such as polyphenols, flavonoids, and betalains. The polyphenolic compounds in fruits and vegetables are responsible for their antioxidant effects [8]. The highest amount of phenolic content is found in the beets’ peel, followed by the crown, and then the flesh [9]. Beets contain a number of flavonoids, among which are quercetin (0.13 mg/100 g) and luetolin (0.37 mg/100 g) [10]. Flavonoids are powerful antioxidants with anti-inflammatory and immune system benefits. Diets rich in flavonoid-containing foods are sometimes associated with the prevention of cancer, neurodegenerative disease and cardiovascular disease [11]. 

Beets also contain p-coumaric acid, feruloylamaranthin, and ferulic acid [9], where the antioxidant properties of p-coumaric acid helps in lowering oxidative stress and inflammation [12]. Feruloylamaranthin helps in lowering inflammation and cholesterol levels and facilitates weight loss [13]. Ferulic acid has significant antioxidant and anti-inflammatory properties [14]. Beets also contain polysaccharides, including galacturonan, glucose polysaccharides (28–39% as cellulose [15] and high-methylated pectin ~70%). Cellulose provides structural support to the cell wall in plants, and when it is consumed, it serves as energy storage and aids in gastrointestinal function [16]. High-methylated pectin helps to improve blood sugar, reduce fat levels, and facilitate weight loss and the digestion of food [17]. Thus, beets are a good source of many nutritional and medicinal compounds.

Betalains, the major phytochemicals found in beets, are water-soluble nitrogen-containing pigments that impart the red–purple natural color to the beet. They are classified into two structure-based groups: the red violet betacyanins and the yellow betaxanthins [18]. Betalain exhibits antioxidant, anti-inflammatory and antiviral properties [19,20]. Betalain, in its form as betacyanin, contains a cyclic amine group and a partly glucosized phenolic group that is responsible for its strong antioxidant effects [21]. The disadvantage of betalains is that they have a relatively low bioavailability, which limits their physiological potential [22]. 

The application of ultraviolet-visible (UV-Vis), infrared (IR) and nuclear magnetic resonance (NMR) spectroscopic methods in combination with chromatography are among the key analytical techniques to profile metabolites in natural products [23]. One of the purposes for such analysis is to identify active medicinal compounds among these metabolites. NMR spectroscopy with its numerous approaches and nondestructive properties has been shown to be an essential tool in plants’ biochemical and medicinal research.

In this study, several techniques, including GC-MS, LC-MS, and other spectroscopy methods such as UV-Vis and NMR, were utilized to demonstrate how the collaborative use of these tools offers an efficient approach to profile metabolites in both the organic and aqueous extracts of dried beetroot. The multi-analytical-tool approach indicated the advantages that each tool provides to yield a more comprehensive profiling of metabolites in the extracts of a natural product. When such an approach is utilized with the organic phase extract of beetroots, it reveals the presence of some bioactive compounds, such as dehydroabietic acid [24], which has pronounced antiviral, antitumor [25], wound healing, and antibacterial activities [26]. The use of 1D NMR selective TOCSY, which requires less instrument time and is more sensitive than 2D TOCSY, enabled the quick profiling of metabolites that are not that prevalent, such as the unsaturated fatty acids in the organic phase extract whose percentage, as shown in the results section, is significantly less than the aqueous phase. The ^31^P NMR data (Appendix A) indicated that orthophosphate monoesters and orthophosphate diesters exist in both aqueous and organic phases. Thus, the adopted approach in this report can facilitate the discovery of more phytochemical and medicinal metabolites. The approach is also essential prior to the investigation of the use of nanoparticles to selectively adsorb certain metabolites from the aqueous and organic phase extracts, which is ongoing in our lab.

## 2. Materials and Methods

### 2.1. Chemicals and Reagents

Fresh beetroots were obtained from a supermarket in Greensboro, NC, USA. Methanol, HPLC grade chloroform and hexane were purchased from Fisher-Scientific, Waltham, MA, USA. Deuterated chloroform (CDCl_3_) with 1% *v*/*v* 3-(Trimethylsilyl)propionic-2, -3-(Trimethylsilyl)prop-98 atom % D (TSP) was obtained from Acros Organics, Morris Plains, NJ, USA. Optima grade water and Acetonitrile for LCMS were obtained from Fisher-Scientific, Waltham, MA, USA. Sodium phosphate dibasic (Na_2_HPO_4_, 99.0%) was obtained from Alfa Aesar, Tokyo, Japan; sodium phosphate monobasic (NaH_2_PO_4_, 99.0%), and sodium azide 99% extra pure were obtained from Acros Organics, Morris Plains, NJ, USA. L-lysine (C_6_H_14_N_2_O_2_, 98.5%), L-leucine (C_6_H_13_NO_2_, 98.5%), L-histidine (C_6_H_9_N_3_O_2_, 98.5%), L-phenylalanine (C_9_H_11_NO_2_, 98.5%) and sucrose (C_12_H_22_O_11_) were purchased from Fisher BioReagents, Pittsburgh, PA, USA.

### 2.2. Sample Preparation

Beets were peeled and diced into small pieces on a watch glass and weighed. The diced beets were dried in a convective air oven (ThermoScientific Heratherm OGS 180, Waltham, MA, USA) for 24 h at 53 °C [27]. Drying continued at 53 °C for an additional 2 h until a constant mass was reached. 

### 2.3. Extraction 

Dried samples were ground using a coffee grinder to obtain a fine powder. About 0.05 g of the powdered beets were placed in a microcentrifuge tube equipped with glass beads; 0.5 mL aqueous methanol (66%/34% *v*/*v*) and 0.5 mL chloroform were added. The sample was then placed in a BioSpec Mini BeadBeater16 at 3400 rpm for 10 min and then centrifuged at 14.8 × 10^3^ rpm for 10 min at 20 °C using Legend Micro 2LR centrifuge, Fisher-Scientific, Waltham, MA, USA. 

Separation: After the centrifugation, three separate layers resulted from the extraction: polar (top), non-polar (bottom) and a solid layer in between. The top phase contained polar compounds dissolved in aqueous methanol. The bottom phase contained the non-polar compounds dissolved in chloroform. The top polar and bottom non-polar phases were micro-pipetted into separate microcentrifuge tubes and placed along the microcentrifuge tube containing the middle layer into a Savant SpeedVac SPD1030 Integrated Vacuum Concentrator, Fisher-Scientific, Waltham, MA, USA, at room temperature and a pressure of 8 torr for 4–6 h. The amounts of the three dried phases were then determined. 

### 2.4. Instrumental Analysis

UV-Vis: A small portion of the polar extract was dissolved in 4 mL of sodium phosphate buffer. The UV-VIS absorption spectrum was recorded in the range 250–750 nm on a Shimadzu UV-2501 PC Spectrophotometer in quartz cuvettes in the absorption mode, where sodium phosphate buffer was the reference. 

GC-MS: The methyl esterification of the non-polar phase was carried out using the standard method [28]. Gas Chromatography-Mass Spectrometry (GC-MS) data was collected using an Agilent 7890 B GC system featuring a 7693 A autosampler provided by Agilent. Mass detection was accomplished utilizing the 5977 GC/MSD instrument.. For the GC system, an Agilent GC HP-5 capillary column (30.0 m length, 0.25 mm i.d., 0.10 µm film thickness) was used. The temperature program was set up starting at 100 °C for 3 min and programmed to increase to 200 °C for 1 min, and ramped up to 250 °C at 10 °C/min, and remained at 250 °C for 10 min for a total program time of 15 min. Both the injector and detector temperatures were 250 °C and Helium gas was used as the carrier gas. The injection volume was 2 µL. Ionization was conducted by electron impact (EI) and Ionization energy; an IE of 70 eV was used for the mass spectroscopy detector with a source temperature at 230 °C and transfer line temperature of 250 °C. The scan range of the fragments was set to 40–500 amu. The fragmentation pattern in the experimental mass spectra were compared with the NIST20.L Mass Spectral Library. Data were acquired using GC-MS acquisition software (mass hunter qualitative analysis 10.0).

LC-MS: Liquid chromatography separation of the metabolites in the polar phase was performed on a Thermo Fisher Q Exactive Plus Mass Spectrometer, Fisher-Scientific, Waltham, MA, USA coupled to a Waters Acquity Ultra-Performance Liquid Chromatography system using a Waters Acquity HSS (100 mm × 2.1 mm) column, Waters Corporation, Milford, MA, USA. A 3 µL sample injection was eluted at a flow rate of 0.4 mL/min from the column employing a binary solvent system comprising 0.1% formic acid in water (designated as mobile phase A) and 0.1% formic acid in acetonitrile (designated as mobile phase B). The gradient program was as follows: 0–1 min 100% A, 1–11 min 100% A–0% A, 11–13.1 min 0% A–100% A, and 13.1–15 min 100% A. The LC eluent was directed into, without splitting, a Thermo Fisher Q Exactive Plus mass spectrometer, Fisher-Scientific, Waltham, MA, USA, fitted with a Heated Electrospray ion (HESI) source, and the MS was operated using the following parameters: source, heated electrospray ionization (HESI); polarity, Pos/Neg switching; capillary voltage, 2500 V; capillary temperature, 262.5 °C; sheath gas 50 L/min; auxiliary gas and spare gas set at 12.5 and 2.5 units, respectively, and the heater temperature was set at 425 °C. The LC-MS data were acquired over a scan range of 75–1125 amu. 

NMR: Deuterated chloroform (CDCl_3_) with 1% TSP as internal reference (0 ppm) was used to dissolve the non-polar phase. The polar phase was dissolved in a sodium phosphate buffer (pH 7.4) that contained TSP and 0.5% sodium azide in 90% water/10% D_2_O. The NMR spectra were acquired on a Bruker Ascend 400 MHz spectrometer at 25 °C. Standard 1D NOESY pulse sequence (with HDO presat pulse for the polar phase) was used to acquire the ^1^H spectrum. One-dimensional selective TOCSY data were collected using the homonuclear Hartman–Hahn (HOHAHA) transfer pulse sequence, where the MLEV17 sequence was used for mixing and the selective excitation was obtained using a shaped pulse and Z-filter [29] with varying mixing times (0.03, 0.08, 0.12 s); the number of scans was set to 256. The data were processed with LB of 0.1–1.0 Hz. Two-dimensional NMR correlation spectroscopy (COSY) spectra were acquired using standard non-phase sensitive sequence (2D homo-nuclear shift correlation [30]). Data were collected with 2KX256 data points matrix, then zero-filled to 2KX1K data points matrix. Total COSY (TOCSY) 2D spectra were acquired using phase sensitive homonuclear Hartman–Hahn (HOHAHA) transfer using MLEV17 sequence for mixing [31] with 2KX256 data points, and zero filled to 2KX1K data point matrix.^1^H- ^13^C single quantum correlation (SQC) data were acquired using the phase sensitive, 2D H-1/X correlation via double inept transfer using the sensitivity improvement pulse sequence [32]. Data were acquired in 2KX256 data points and zero-filled to 2KX1K data points. 

## 3. Results and Discussion

### 3.1. Extraction

Based on four trials, the extraction data indicate that the average percentages of the aqueous and organic phases are 33.50 and 3.05, respectively. The average percentage of the middle solid layer that contains compounds that are not soluble in water/methanol or in chloroform is 63.45. 

### 3.2. Aqueous (polar) Phase

The UV-Vis spectrum of the aqueous phase in Figure 1 shows the two signals corresponding to the two forms of the betalain pigment: the red–purple betacyanins at 538 nm and the yellow betaxanthins at 484 nm [18]. The relative intensity of the two signals in Figure 1 is consistent with the higher composition of the red–purple betacyanins relative to that of the yellow betaxanthins [18]. 

Appendix A shows the MS fragmentation spectrum of the compounds in the aqueous extract of beetroot powder in positive polarity mode. The major difference in ionizability of the identified compounds in the polar phase (Table 1) rendered the intensity of the signals in the LC-MS spectrum to be widely varied (Appendix A). Most identified compounds by LC-MS data were found by searching for their corresponding ions in the positive mode.

Table 1 lists the identified compounds in the aqueous extract from the LC-MS results (Appendix A) and the standards when applicable. Fourteen amino acids were identified. The NMR signals corresponding to some of these amino acids were also identified, as shown below (Figure 2).

Figure 2 shows the ^1^H NMR spectrum of the aqueous extract that indicates the significant variance in the composition of the different compounds, where the intensity of the signals corresponding to the aromatic compounds is much less than the intensity of the signals corresponding to other compounds such as sugars and amino acids. Insert A in Figure 3 shows an expansion of the up-field region of the 1D NMR spectrum between 0.5 and 3.2 ppm, and insert B shows the region of the spectrum where aromatic compounds resonate.

Figure 3 is the section of the 2D TOCSY spectrum of the aqueous extract, which exhibits spin coupling between C_1_H of sucrose and the other protons in it. The spectrum also shows the cross peaks corresponding to coupling of C_1_H proton in the α- and β- forms of D-glucopyranose to other protons in them. Appendix A lists the chemical shift of the identified protons in the two sugars and the corresponding literature chemical shift values [33]. It is interesting to note that a part of the betacyanins dye is a D-glucopyranose-like six-member ring [18], which means that some of the observed couplings in Figure 4 could belong to the betacyanins pigment. The relative intensity of C_1_H signals of α- and β-forms of D-glucoyranose in Figure 3 is consistent with the literature, indicating that the β form is more abundant than the α form [34]. Figure 4 is the section of 2D HSQC spectrum that shows the ^1^H/^13^C correlation corresponding to sucrose; the chemical shift values of the sucrose ^13^C signals are listed in Appendix A along with the corresponding literature values. Figure 5 shows the section of COSY spectrum that exhibits the spin coupling between the two methyl groups of valine and C_2_H proton. Appendix A show the sections of COSY spectrum that exhibit spin coupling corresponding to the isoleucine. Appendix A shows combined sections from the COSY spectrum of the aqueous phase, which display the coupling corresponding to leucine. Appendix A lists the chemical shift of the identified protons in leucine, isoleucine and valine along with the literature chemical shift. It Is interesting to note the similarity between the chemical shift of ^1^H and ^13^C signals of the all unambiguously identified compounds and the literature data (Appendix A ), where the literature reported chemical shift values are from the spectra of pure compounds, indicating that there is no significant matrix effect on the chemical shift. Many of the bioactive compounds in beetroots, such as flavonoids and p-coumaric acid, are aromatic compounds. The 1D NMR spectrum in Figure 2 and insert B show that the intensity of the signals corresponding to the aromatic compounds is much lower than that of the sugar signals, which are in the range of 3 to 5.5 ppm. This indicates that the composition of the aromatic compounds is significantly lower than that of sugars, which made detecting the spin systems corresponding to them, including the pigment’s signals, more difficult, even while using the 1D selective TOCSY technique, which is more sensitive than 2D NMR techniques. Insert A in Figure 2 shows the significant overlap of signals in the ppm range of 0.90 to 1.05 ppm, where the methyl groups usually resonate. Figure 5 and Appendix A show how 2D NMR experiments can be utilized efficiently to identify some of the molecules whose NMR signals show significant overlap. The figures also indicate how NMR techniques can efficiently complement LC-MS data.

### 3.3. Organic Phase

Figure 6 shows the GC-MS chromatogram of the organic extract of dried beetroot after chemically converting the fatty acids to methyl esters; the retention time of the eluted compounds ranges between 6.50–25.50 min. For the identification of fatty acids methyl esters (FAME), and other compounds in the organic phase, retention times and the MS ionization spectra of the experimental data were compared with the corresponding spectra from the NIST20 library [28]. For example, Appendix A shows the MS experimental and NIST20 library spectra of 9,12-Octadecadienoic acid methyl ester (RT 10.359 min). Appendix A show the matching MS spectra for Linolenic acid, Oleic acid, Stearic acid, and Palmitic acid, respectively. The similarity between the fragments in the two MS spectra was reported as matching/similarity score. Table 2 lists the identified compounds in the organic phase along with their molecular formula and their corresponding retention times, base peak signal-to-noise ratio, base peak area, and the similarity scores being 88% and above. Figure 7a shows the ^1^H NMR spectrum of the organic phase, while traces 7b, c and d are the selective TOCSY traces that were used to identify the spin coupling in linolenic, linoleic and oleic acids [35], respectively, where certain signals were selectively excited in each trace. The broad peak at 5.36 ppm corresponds to the olefinic protons (protons 9, 10, 12, 13, 15, and 16) in linolenic acid, (protons 9, 10, 12, and 13) in linoleic acids and protons 9 and 10 of oleic acid. The multiplet at 2.77 ppm corresponds to the bis-allylic CH_2_ protons 11 and 14 in linolenic acid, and protons 11 in linoleic acid. The multiplet at 2.31 ppm (designated as peak 2) corresponds to the CH_2_ group adjacent to the carboxylic group in the three fatty acids. The multiplet at 1.60 ppm (designated as peak 3) corresponds to the CH_2_ group next to the CH_2_ group labeled as two in the three fatty acids. The multiplet at 2.06 ppm corresponds to the allylic CH_2_ in the three fatty acids: protons 8 and 17 in linolenic acid, 8 and 14 in linoleic acid and 8 and 11 in oleic acid. The intense peak at 1.26 corresponds to all other CH_2_ groups in the three and the other saturated fatty acids. The peak at 0.88 ppm corresponds to the terminal methyl in all fatty acids. When the peak at 1.26 ppm (the overlaping CH_2_ groups) was selectively excited (trace 7b), the predicted spin coupling correlations in the three fatty acids were observed according to the above-mentioned assignments. Similarly, when the peaks at 2.06 ppm (allilic CH_2_(s)) and 0.88 ppm (terminal methyl) were selectively excited (traces 7c and d, respectively), the predicted spin coupling correlations in the three fatty acids, according to the above-mentioned assignments, were observed. Appendix A shows the chemical shift of the assigned protons in fattty acids, where the reported chemical shift values in the literature are identical [35]. The selective TOCSY experiment proves to be useful in assigning signals belonging to unsaturated fatty acids. Thus, in a mixture of chemical compounds similar to the aqueous or organic phase of the extracts of a natural product, selective TOCSY is a good tool to identify metabolites when these metabolites have unique chemical properties such as unsaturated carbons or aromatic ring. Still, one has to be aware of the signal overlaps that prohibit the identification of all the peaks in the selective TOCSY traces, like the peak at 2.90 ppm in trace 7b. Appendix A shows the proton-decoupled ^31^P spectra of the aqueous phase extract in D_2_O (A) and the organic phase extract in CDCl_3_ (B). The chemical shift of ^31^P peaks in both media falls between 0 and 6.5 ppm. The peaks were then assigned to orthophosphate monoester and diester [36]. Thus, both mono and diester exist in the two phases. The difference in the chemical shifts of the phosphorus signals in the two phases has to do with R groups being different and the solvent systems used (D_2_O and CDCl_3_).

## 4. Conclusions

The current report is an evidence of the efficacy of combining chromatography and spectroscopy data in profiling metabolites and bioactive compounds in beetroot specifically, and, more generally, in natural product research. The report shows how NMR in its multifaceted tools, such as 1D of different nuclei and the variety of homo-nuclear and hetero-nuclear 2D experiments, can efficiently complement LC/MS and GC/MS. For example, the metabolites detected by LC/MS data (Table 1) include glucose, while 2D TOCSY data (Figure 3) showed the spin coupling corresponding to the two forms of glucose α and β. The projection along second dimension in Figure 3 shows the relative abundance of these two glucose forms. The methyl groups of amino acids resonate up-field from most signals. Albeit the significant signal overlap, Figure 5 and Appendix A demonstrate how 2D NMR can be utilized to assign overlapping signals. The chemical shift of the ^1^H, ^13^C and ^31^P signals that correspond to the unambiguously identified metabolites in both aqueous and organic phases (Appendix A) match very well with the chemical shift of the corresponding metabolites in the literature, where these chemical shifts were obtained from the spectra of pure metabolites. This indicates that the matrix effect on the chemical shift is negligible, which means that the literature chemical shift values can be effectively used in identifying metabolites in the NMR spectra of natural products. The low abundance of the aromatic metabolites in the aqueous phase (Figure 2), such as polyphenols and the two forms of the pigments, prohibited the unambiguous assignment of these compounds, even by using 1D selective TOCSY experiments. UV-vis (Figure 1) and LC/MS (Appendix A and Table 1) were the effective tools to identify such metabolites, indicating the collaborative nature of the tools used in this report. The ^31^P NMR spectrum of the aqueous and organic phases indicate that ortho monoester and diester phosphates are in both phases. The difference in polarity of the two phases suggests that the R groups in the mono and dieters of the organic phase are longer, which makes these esters more hydrophobic. Finally, the profiling of metabolites and the analytical tools used in this report prove to be valuable in the ongoing investigation in our lab to study the use of nanoparticles to selectively separate these metabolites as they are selectively adsorbed on the nanoparticles.

## Figures and Tables

**Figure 1 foods-12-03510-f001:**
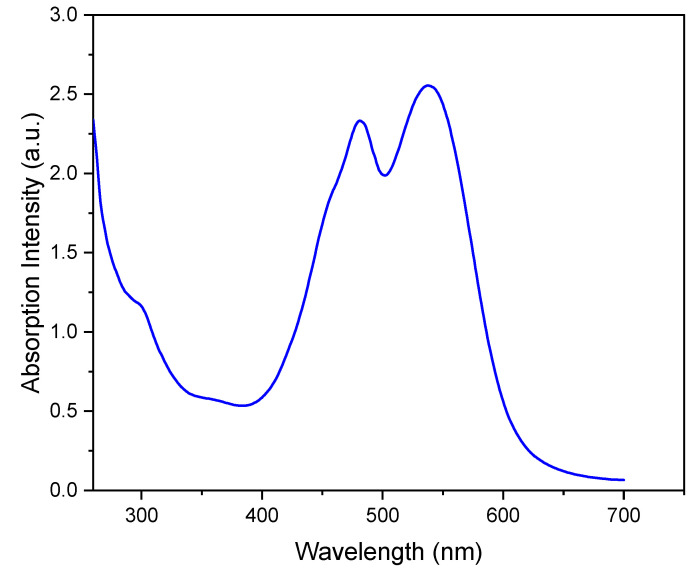
UV-vis spectrum of aqueous beetroot extract.

**Figure 2 foods-12-03510-f002:**
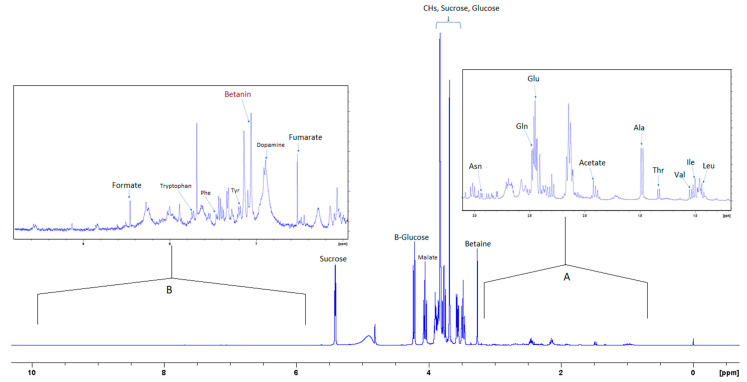
400 MHz ^1^H-NMR spectrum of aqueous extract in phosphate/D_2_O buffer with 0.5% 3-(trimethylsilyl)propionic-2,2,3,3-d_4_ acid sodium salt (TSP). Inset (**A**) shows 10× magnification of the upfield region; (**B**) shows 20× magnification of the downfield region.

**Figure 3 foods-12-03510-f003:**
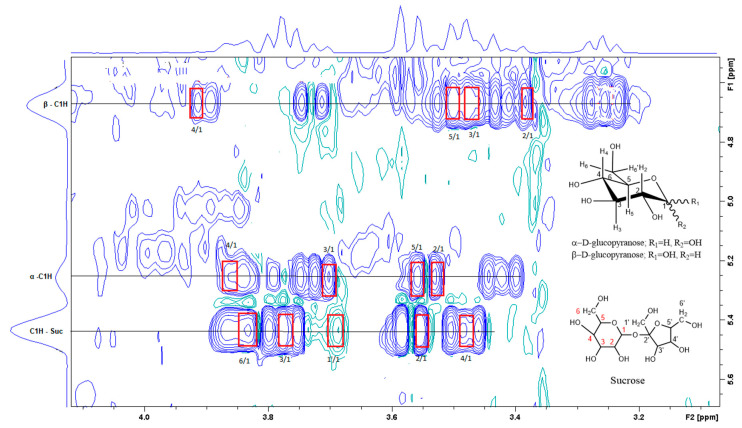
A section of the TOCSY spectrum of aqueous extract shows the spin coupling between C_1_H of sucrose and of glucose and the corresponding protons.

**Figure 4 foods-12-03510-f004:**
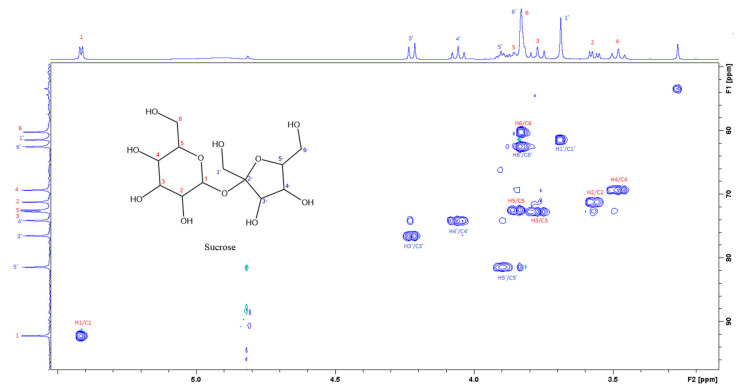
The section of the 400 MHz 2D-HSQC spectrum of aqueous extract shows the correlation between ^1^H and ^13^C of sucrose.

**Figure 5 foods-12-03510-f005:**
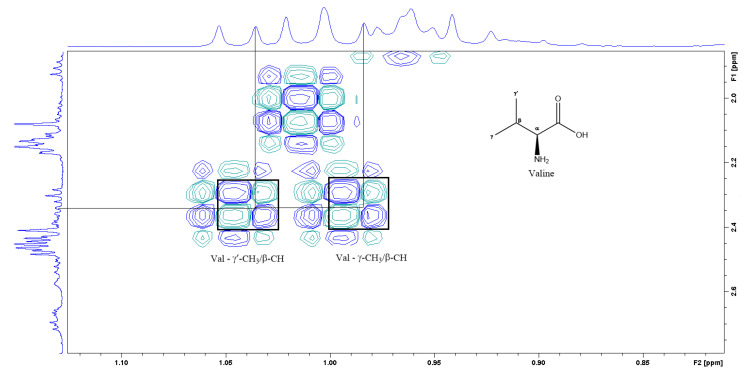
A section of the COSY spectrum showing the spin coupling corresponding to Valine.

**Figure 6 foods-12-03510-f006:**
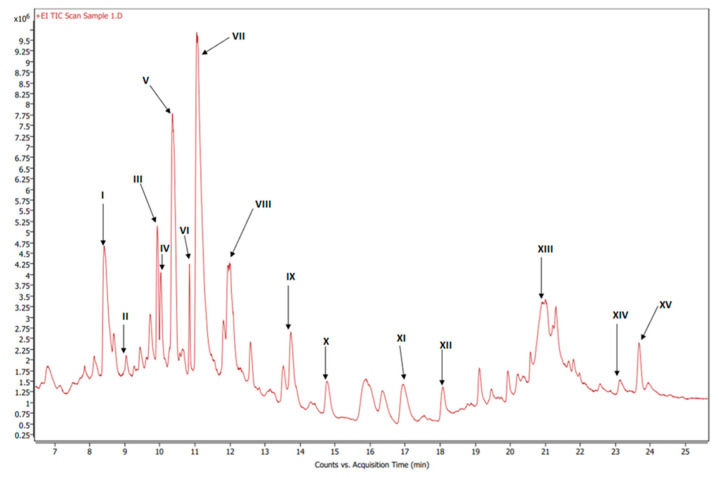
GC-MS chromatogram of esterified organic extract of dried beets.

**Figure 7 foods-12-03510-f007:**
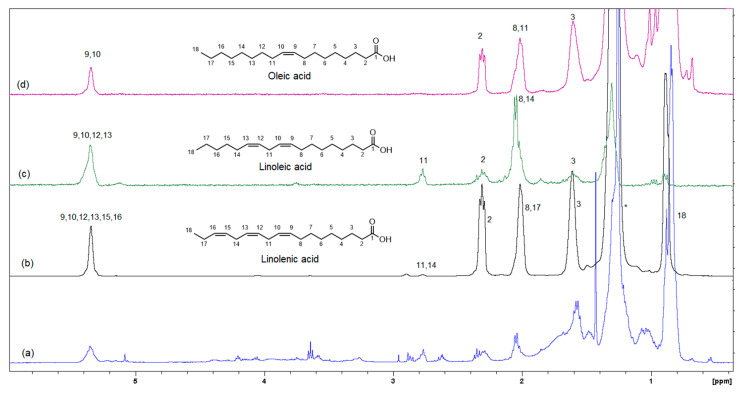
Selective TOCSY data showing the spin coupling correlation in linolenic, linoleic and oleic acids (mixing time 0.080 s). (**a**) Shows the ^1^H-NMR spectrum of the organic phase, while traces (**b**–**d**) are the selective TOCSY traces that were used to identify the spin coupling in linolenic, linoleic and oleic acids, respectively.

**Table 1 foods-12-03510-t001:** Identified compounds in the aqueous extract using LC-MS in positive polarity mode.

Compound	Molecular Formula	Measured [M + H]^+^ (*m*/*z*)	Measured Rt (min)	Standard [M + H]^+^ (*m*/*z*)	Rt (min) of Standard	Confirmed by NMR
Lysine	C_6_H_14_N_2_O_2_	147.1131	0.58	147.1126	NA	No
Histidine	C_6_H_9_N_3_O_2_	156.0771	0.61	156.0765	0.61	No
Arginine	C_6_H_14_N_4_O_2_	175.1193	0.64	N/A	NA	No
Threonine	C_4_H_9_NO_3_	120.0659	0.68	N/A	NA	No
Glutamic acid	C_5_H_9_NO_4_	148.0606	0.68	N/A	NA	No
Valine	C_5_H_11_NO_2_	118.0867	0.71	N/A	NA	Yes
Proline	C_5_H_9_NO_2_	116.0710	0.77	N/A	NA	No
Sucrose	C_12_H_22_O_11_	343.1239	0.98	343.1229	0.98	Yes
Glucose	C_6_H_12_O_6_	181.0710	0.77	N/A	NA	Yes
Methionine	C_5_H_11_NO_2_S	150.0587	1.25	N/A	NA	No
Leucine	C_6_H_13_NO_2_	132.1022	2.53	132.1018	2.49	Yes
Isoleucine	C_6_H_13_NO_2_	132.1023	2.68	N/A	NA	Yes
Tyrosine	C_9_H_11_NO_3_	182.0816	2.68	N/A	NA	No
Betacyanin	C_24_H_26_N_2_O_13_	551.1520	2.96	N/A	NA	Yes
Phenylalanine	C_9_H_11_NO_2_	166.0867	3.02	166.0860	3.02	No
Tryptophan	C_11_H_12_N_2_O_2_	205.0975	3.48	N/A	NA	No
Riboflavin	C_17_H_20_N_4_O_6_	377.1442	3.85	N/A	NA	No
Betaxanthin	C_18_H_18_N_2_O_6_	359.1247	4.11	N/A	NA	Yes
Theanine	C_7_H_14_N_2_O_3_	175.1078	13.82	N/A	NA	No

**Table 2 foods-12-03510-t002:** MS identification of compounds from esterified organic extract of dried beets. The methyl esters of the fatty acids are presented in a bold label.

Label	RT (min)	Base Peak S/N Ratio	Base Peak Area	Compound Name	Formula	Match/Similarity Score (%)
**I**	8.41	9.73 × 10^2^	6.15 × 10^6^	Hexadecanoic acid, methyl ester	C_17_H_34_O_2_	93.9
II	9.33	3.88 × 10^1^	2.69 × 10^5^	3-Methylbenzoic acid, 2,5-dichlorophenyl ester	C_14_H_10_Cl_2_O_2_	88.7
**III**	9.93	1.0 × 10^3^	2.69 × 10^6^	Methyl stearate	C_19_H_38_O_2_	96.5
**IV**	10.03	2.35 × 10^2^	7.04 × 10^5^	9-Octadecenoic acid, methyl ester, (E)-	C_19_H_36_O_2_	90.7
**V**	10.36	3.42 × 10^2^	3.70 × 10^6^	9,12-Octadecadienoic acid (Z,Z)-, methyl ester	C_19_H_34_O_2_	91.9
**VI**	10.85	3.08 × 10^2^	6.45 × 10^5^	9,12,15-Octadecatrienoic acid, methyl ester, (Z,Z,Z)-	C_19_H_32_O_2_	96.4
VII	11.99	3.58 × 10^3^	2.42 × 10^7^	Dibutyl phthalate	C_16_H_22_O_4_	91.4
VIII	12.59	6.44 × 10^1^	1.04 × 10^6^	Pentacosane	C_25_H_52_	91.7
IX	13.74	1.66 × 10^2^	1.57 × 10^6^	n-Hexadecanoic acid	C_16_H_32_O_2_	92.7
X	14.77	4.59 × 10^1^	1.07 × 10^6^	Octacosane	C_28_H_58_	90.1
XI	16.95	8.14 × 10^1^	1.31 × 10^6^	Octadecanoic acid	C_18_H_36_O_2_	92.6
XII	19.46	2.31 × 10^2^	4.92 × 10^5^	3,5-di-tert-Butyl-4-hydroxyphenylpropionic acid	C_17_H_26_O_3_	90.0
XIII	21.21	1.50 × 10^2^	1.02 × 10^6^	Oxybis(propane-1,2-diyl) dibenzoate	C_20_H_22_O_5_	90.7
XIV	23.14	2.38 × 10^2^	1.20 × 10^6^	Diethylene glycol dibenzoate	C_18_H_18_O_5_	97.2
XV	23.94	1.87 × 10^2^	3.80 × 10^5^	Dehydroabietic acid	C_20_H_28_O_2_	88.1

## Data Availability

Between the data in the figures in the manuscript and the Appendix A most data are presented. Any other data that the science community shows interest in can be shared upon request from the corresponding author.

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
