# Peer review of "Multipronged Approach to Profiling Metabolites in Beta vulgaris L. Dried Pulp Extracts Using Chromatography, NMR and Other Spectroscopy Methods"

_foods, 2023, doi:10.3390/foods12183510_

Round 1
Reviewer 1 Report
Manuscript Number: foods-2561938
Title: Multipronged Approach to Profiling Metabolites in Beta Vulgaris L Dried Pulp Extracts Using Chromatography, NMR, and Other Spectroscopy Methods
Comments to the authors
The authors propose employing chromatographic, spectroscopic, and NMR techniques for the evaluation of metabolite profiles in beetroot dried pulp extracts. The topic addressed in this study appears to be of interest to the journal's readers. However, there are several points in this manuscript that are not clearly explained and require further clarification. Therefore, I recommend that this manuscript undergo major revisions before it can be considered for publication.
To submit the revised version of this manuscript, please consider the following points:
1) Provide a detailed description of the novelty of this work in the Introduction section.
2) Correct typographical errors and spacing issues throughout the entire manuscript. Additionally, ensure that all text is justified.
3) Please, Replace "4.8X103" with "4.8 × 103".
4) Incorporate the Instrumental section within the Materials and Methods section.
5) While this research employs various analytical techniques relevant for characterizing beetroot dried pulp extracts, the manuscript looks crowded with figures and tables, evidencing a lack of comprehensive discussion of the obtained data. Therefore, it is advised to enhance the Results and Discussion section.
6) As per the new structure of sections, number the subsections in the Results and Discussion section as follows: 3.1, 3.2, 3.3, ...
7) Ensure that the quality and sharpness of all figures match that of Figure 4.
8) Highlight the advantages of this study compared to previous similar studies within the Results and Discussion section.
9) Modify the Conclusions section to emphasize the main findings of this study, as the current presentation is more suited for a technical report than a scientific article.
10) Thoroughly review the reference style to prevent errors introduced automatically by the software.
Once these revisions have been made, the manuscript will be better aligned with the standards required for publication.
Author Response
1) Provide a detailed description of the novelty of this work in the Introduction section
A paragraph is added in the introduction part to emphasize the novelty of the work. Also the conclusion part has been changed to make it according to the results part and to describe the novelty of the research.
2) Correct typographical errors and spacing issues throughout the entire Manuscript. Additionally, ensure that all text is justified.
Both items were addressed
3) Please, Replace "4.8X103" with "4.8 × 103".
Done
4) Incorporate the Instrumental section within the Materials and Methods section
Done
5) While this research employs various analytical techniques relevant for characterizing beetroot dried pulp extracts, the manuscript looks crowded with figures and tables, evidencing a lack of comprehensive discussion of the obtained data. Therefore, it is advised to enhance the Results and Discussion section
The results part is made more focused by moving two figures to the supplementary material part. Description of some of the data was changed and more information were added to give a clearer description to the data leading to better explanation of the results part.
6) As per the new structure of sections, number the subsections in the Results and Discussion section as follows: 3.1, 3.2, 3.3, ...
Done
7) Ensure that the quality and sharpness of all figures match that of Figure 4.
Done
8) Highlight the advantages of this study compared to previous similar studies within the Results and Discussion section. Chromatography/MS and NMR analytical methods have been shown to be the right combination for the profiling metabolites in natural products: Avenges of chromatography/MS: Much lower detection limits, Sensitivity
The conclusion part was totally changed to address the points raided by the editors
9) Modify the Conclusions section to emphasize the main findings of this study, as the current presentation is more suited for a technical report than a scientific article
As mentioned above and to to address point 8 of the reviewer, the conclusion part was changed to address the concern about the conclusion section
10) Thoroughly review the reference style to prevent errors introduced automatically by the software.
Once these revisions have been made, the manuscript will be better aligned with the standards required for publication.
Done
Reviewer 2 Report
In the manuscript”
Multipronged Approach to Profiling Metabolites in Beta Vulgaris L Dried Pulp Extracts Using Chromatography, NMR, and Other Spectroscopy Methods” Joshua et.al. performed a profiling of metabolites. The work was carried out nicely, but I have few comments for improvement before recommending for publication.
1. English editing is required throughout the manuscript. For example, in abstract It should be “to profile metabolites in the extracts of beetroot”. Metabolites are natural products and those come from extracts of beet root.
2. What does author mean by “Drying continued until a constant mass was reached”?
3. Does long drying time period degrade sensitive molecules?
4. Label the maximum possible NMR peaks in Figure 3. Make a square figure with expanded region for labeling or annotation.
natural products or metabolites are in the extracts of beet root.
Please be consistent throughout the manuscript.
Author Response
- English editing is required throughout the manuscript. For example, in abstract It should be “to profile metabolites in the extracts of beetroot”. Metabolites are natural products and those come from extracts of beet root.
The reviewer's comment was addressed. Now through the manuscript we used extracts of the beetroot.
- What does author mean by “Drying continued until a constant mass was reached”?
The authors followed the literature procedure to dry the powder, which is drying it for 24 hours. The mass of the powder was checked after the 24 hours drying, and then the powder was put back in the oven for extra two hours to ensure a constant-mass drying.
- Does long drying time period degrade sensitive molecules?
The slow drying at low temperature (53 C) was followed from the literature to prevent any degradation of any molecules in the natural product.
- Label the maximum possible NMR peaks in Figure 3. Make a square figure with expanded region for labeling or annotation.
Figure 3 was changed in a way to label as many peaks as possible. This is by the way a good suggestion by the reviewer because it made it easier to obtain information from the figure
Round 2
Reviewer 1 Report
Manuscript Number: foods-2561938
Title: Multipronged Approach to Profiling Metabolites in Beta Vulgaris L Dried Pulp Extracts Using Chromatography, NMR, and Other Spectroscopy Methods
Comments to the authors
The authors have addressed most reviewer's comments, yet missed key points needed for manuscript quality improvement. While some previous suggestions, especially regarding the result discussion, remain unaddressed, they must be reviewed. On the other side, the lengthy Conclusions section could divert the reader's focus. It's vital to arrive at concise and forceful conclusions that summarize the investigation's main findings and underscore their relevance and contribution to the field. Additionally, ensure uniform justification and alignment of the entire manuscript's text to a consistent margin.
Author Response
In the second revision of the manuscript titled “Multipronged Approach to Profiling Metabolites in Beta Vulgaris L Dried Pulp Extracts Using Chromatography, NMR, and Other Spectroscopy Methods” we have addressed the requested changes by the reviewer 1. The major changes in the manuscript were put in red font to make it easier for the reviewer to see. The reviewer suggested revising the results and discussion part since it looked crowded. Five figures and three tables were moved from the results and discussion to the supplementary material. The discussion now focuses on the main points learned from the data in the figures and tables. The reviewer also commented on the length of the conclusion part. The conclusion part is now significantly shorter where it focuses, among other things, on the advantages of the collaborative use of the analytical tools, especially 1D and 2D NMR, that were utilized in the report to profile metabolites in beetroot. Such collaborative approach is one of the main contribution of this report to the field.